# Peptides to Overcome the Limitations of Current Anticancer and Antimicrobial Nanotherapies

**DOI:** 10.3390/pharmaceutics14061235

**Published:** 2022-06-10

**Authors:** Valentina Del Genio, Rosa Bellavita, Annarita Falanga, Katel Hervé-Aubert, Igor Chourpa, Stefania Galdiero

**Affiliations:** 1Department of Pharmacy, University of Naples “Federico II”, Via Domenico Montesano 49, 80138 Naples, Italy; valentina.delgenio@unina.it (V.D.G.); rosa.bellavita@unina.it (R.B.); 2EA 6295 Nanomédicaments et Nanosondes, University of Tours, UFR Pharmacie, 31 Avenue Monge, 37200 Tours, France; katel.herve@univ-tours.fr; 3Department of Agricultural Science, University of Naples “Federico II”, Via Università 100, 80055 Naples, Italy; annarita.falanga@unina.it

**Keywords:** nanovectors, peptide, anticancer, antimicrobial, drug release

## Abstract

Biomedical research devotes a huge effort to the development of efficient non-viral nanovectors (NV) to improve the effectiveness of standard therapies. NVs should be stable, sustainable and biocompatible and enable controlled and targeted delivery of drugs. With the aim to foster the advancements of such devices, this review reports some recent results applicable to treat two types of pathologies, cancer and microbial infections, aiming to provide guidance in the overall design of personalized nanomedicines and highlight the key role played by peptides in this field. Additionally, future challenges and potential perspectives are illustrated, in the hope of accelerating the translational advances of nanomedicine

## 1. Introduction

Nanotechnology stimulates huge innovation in medical and healthcare treatments and therapies through the control of materials at the nanoscale level. It deals with the preparation of nanosized entities usually ranging from 1 to 100 nm, which compared to bulk materials present unique physicochemical properties that can be implemented in diverse biomedical applications. Thus, nanotechnology is receiving great attention for the achievement of personalized medicine to overcome the limitations of current therapies. Indeed, despite being a constantly progressing subject of medicinal science, drug delivery still represents a crucial challenge [1]. Drug delivery by non-viral nanovectors (NVs) presents several advantages such as the possibility to customize the drug release, solubility, half-life, bioavailability and immunogenicity. The use of nanocarriers such as liposomes, micelle, and nanoparticles of different origin [2,3] has been shown to improve the solubility of drugs and prevent their degradation by enzymes, pH and other factors during blood circulation (Table 1). In addition, tunable size, shape and structure of NVs allow them to reach relevant drug loading capacity. Moreover, being comparable in size to human cell organelles, they can interact with various ligands, both hydrophilic and hydrophobic, target cells and intracellular compartments. 

Undoubtedly, the delivery of a therapeutic agents directly to the target is a critical challenge, important to increase its efficacy while reducing the side effects [4,5]. Conventional chemotherapeutics used to treat cancer present several common limitations such as: (i) low solubility in water due to their hydrophobic character, (ii) lack of selectivity of cancerous cells and (iii) potential to develop multidrug resistance; for instance, some drugs can increase the risk of myocardial infarction, heart attack, stroke and blood clot [6].

Biological distribution is related to the size and shape of NVs. Almost all carrier designs use spherical nanoparticles but recently more groups started to work on anisotropic nanoparticles, which are characterized by a variation in shape with respect to direction. Compared to spherical nanoparticles, anisotropic nanoparticles present higher resistance to non-specific cell elimination [20]. For instance, to fabricate anisotropic nanoparticles, Ben Akiva, et al. [21] synthesized spherical PLGA nanoparticles and then stretched them above the glass transition temperature of PLGA. When nanoparticles were stretched two-fold in one-dimension prolate nanoparticles were obtained, while when they were stretched 1.5-fold in two dimensions oblate ellipsoidal nanoparticles were produced. The anisotropic nanoparticle fluidity or stability was comparable to spherical nanoparticles. Compared with uncoated spherical nanoparticles, anisotropic nanoparticles coated with red blood cell membrane were able to better escape macrophage clearance and showed a superior half-life compared to spherical nanoparticles.

NVs can overcome biological barriers to deliver drugs preferentially to the affected organ and, in this manner, reduce the side effects. Cell membranes and tissue barriers control the movement of the most active molecules and represent a natural defense mechanism which prevents the invasion of external substances and is crucial for cell survival. For that reason, it is critical to find proper ways of membrane translocation and drug delivery to the target site. To design novel drug delivery systems, it is key to take into consideration the different biological barriers starting from tissues to cells, and organelles to ensure an ideal therapeutic index of drugs [22]. The NV role is to prolong blood circulation and penetration through barriers, increase tissue accumulation and enhance cellular uptake while inhibiting the drug efflux. Eventually, it should control intracellular drug distribution till the action site. 

As previously mentioned, endothelial barriers prevent drug extravasation from the bloodstream [23]. In cancers, the existence of nanoscale gaps between vascular endothelial cells allows substances to leak into tumors, via the enhanced permeability and retention (EPR) effect that favors the preferential accumulation of the NVs in the tumor site. However, low interstitial fluid pressure (IFP) and highly cross-linked extracellular matrix (ECM) severely prevent the further penetration and diffusion in tumors. The strong metabolism of tumor cells causes the ECM which is characterized by the large amount of collagen and polysaccharides produced in the tumor site and determines the failure of drugs to spread all over the entire tumor and their accumulation at the edge of it. To reduce their clearance by RES (reticuloendothelial system), the NVs may be coated with a shell of polyethylene glycol (PEG). The PEGylation is the commonly used strategy to avoid adsorption of opsonin proteins on the nanosystem surface (formation of the protein corona [24,25,26]) and to prolong their blood circulation in vivo, till delivery to the target tissues [27]. For drugs targeting brain, there is need to cross the blood–brain barrier (BBB), which is mainly composed of tight junctions between vascular endothelial cells, a capillary basement membrane and a glial membrane formed by astrocytes and pericytes. The junctions are even more complex and tight than those in other vessels; in fact, the role of the BBB is to prevent most substances from entering the brain parenchyma and thus to protect the central nervous system (CNS) [28]. Controlling drug translocation across the BBB is helpful to improve the distribution of drugs. 

NVs offer promise also against microbial infections partially because of their ability to elude existing mechanisms used by drug-resistant bacteria [29]. Furthermore, other bacterial survival mechanisms, such as biofilms, provide impediments to effective therapies and the use of NVs, as alternatives or in synergy with existing antibiotics, is one research area with the highest potential to be effective when conventional antibiotic therapies fail. Interestingly, NV-based antimicrobials may act with different mechanisms: (i) the NVs can act as nanocarriers for the efficient delivery of drugs, (ii) the NVs possess inherent antimicrobial activity on their own and (iii) the NVs may work using a combination of both properties.

The purpose of this review is to highlight the applications of NVs with specific reference to those made of peptides, in two main types of pathologies, cancer on one side and microbial (bacterial/viral) infections on the other side (Figure 1). Development of new therapeutic strategies that improve the quality of life of patients will be the long-term impact of these studies [30,31].

## 2. NVs of Anticancer Drugs

### 2.1. Peptides to Enhance the NVs Potential 

Peptides are versatile materials for the development of NVs. Peptides can be synthesized with different strategies and coupled to NVs to improve their bioavailability, recognition of specific target cells by interacting with molecules/receptors overexpressed on their surface (cell targeting peptides—CTP) and internalization into cells by interacting with membranes (cell penetrating peptides—CPP). Although peptides show high promise for the targeting and intracellular delivery of next-generation nanotherapeutics [32,33], their proteolytic susceptibility is one of the major limitations of their activity in a biological environment [34]. Numerous chemical approaches have been formulated to improve peptide resistance to proteolysis, including N- and C-termini protection, cyclization, backbone changes, incorporation of non-canonical amino acids and conjugation of cargoes [35]. Furthermore, conjugation of peptides to NVs not only will enhance targeting and cell penetration but will also provide some degree of shielding. The peptide conjugation also increases the biocompatibility of the NVs. Below we report the main features and applications of CTP and CPP (Figure 2).

### 2.2. Methodologies Used to Immobilize Peptides on NVs

Peptides can be coupled to NVs with different methodologies such as physical adsorption, electrostatic binding and covalent coupling. Certainly, the preferred binding strategy is the covalent conjugation, which allows the obtainment of highly stable complexes. Nonetheless, peptide absorption or electrostatic binding may represent the key strategy for reversible binding, depending on pH, ionic concentration, temperature or other environment physicochemical parameters [36].

Isothiocyanate (NCS), N-hydroxysuccinimide-esters (NHS-esters) or maleimide are the most used groups for the covalent coupling of primary amines or thiols. Thanks to the ability of thiols to form stable linkages, the selectivity of the reaction and the good yields, this strategy is widely used for binding a peptide to a NV through the thiol (-SH) side chain of a cysteine residue by means of maleimide chemistry, bromoacetylation or vinylsulfone [37]. On the other hand, peptides contain specific and reactive functional groups on side chains available to form covalent couplings with NVs such as (i) e-amino group in lysines, (ii) carboxylic acid in aspartic and glutamic acids and (iii) hydroxyl group in serine and tyrosines. Moreover, among the bio-orthogonal click reactions, the most widely applied is the copper-catalyzed azide-alkyne cycloaddition reaction (CuAAC) [38]. The choice of the bioconjugation procedure depends strictly on physicochemical and biochemical properties of NVs and peptides, the most popular methodologies used for bioconjugation are summarized in Table 2. For instance, factors such as the NV polydispersity, area, porosity and charge are very important for the choice of the binding strategy since the hydrophobicity, charge and site affinity could affect the interaction and thus compromise the stability of the final product.

Clearly, when covalently binding peptides to NVs it is important to confirm retention of activity after the binding. A careful design of the synthetic strategy includes the analysis of the active domain of the peptide which should be left free to perform its activity also when the peptide is bound to the NV; this could be enabled by using the functional groups located the farthest from the active domain for the binding and by adding linkers between the peptide and the NV.

### 2.3. CTP-Decorated Anticancer NVs

Active drug targeting is critical for cancer treatment, where initially only passive targeting, exploiting the EPR effect, has been used. For cancer therapy, nanoparticles with diameters of 10–100 nm exploit the EPR effect for delivery; smaller nanoparticles (sizes less than 1–2 nm) leak from the normal vasculature and damage normal cells while larger nanoparticles (greater than 100 nm) may be cleared from circulation by phagocytes. Different from passive targeting that involves the EPR effect, active targeting relies on the direct interaction between ligands and receptors. 

Active tumor targeting is based on an altered gene and protein expression typical of cancer cells which may be exploited to enhanced drug accumulation at the tumor site [39]. Usually, tumors over-produce molecular components that support their development and metastasis. An overexpression of G-protein-coupled receptors, growth factor receptors, interleukins, transferrin receptors, folate receptors and polysaccharide moieties are frequently observed. 

The most efficacious targeted therapies used up to now involve monoclonal antibodies; nonetheless, several options are being developed. Great efforts are devoted to exploring smaller targeting moieties such as peptides to increase tissue selectivity. In fact, peptides possess a high affinity and selectivity, are easy to synthetize and derivatize and usually have low immunogenicity. 

The interactions between receptors and CTP bound ligands could be applied to targeting cancer tissues since cancer cells express a distinct set of receptors compared to that of normal tissue.

CTP are made of 3–15 amino acids that specifically identify and adhere to tumor cells/vasculature, and thus target tumors and tumor microenvironments. Most widely used CTP are: (i) the RGD motif (Arg-Gly-Asp) which binds to α integrins, and (ii) the NGR motif (Asn-Gly-Arg), which binds to a receptor aminopeptidase on the surface of endothelial cells. RGD and NGR are tumor-targeting peptides and widely exploited to decorate the surface of NVs for the treatment of a range of tumor models.

RGD motif is one of the most studied receptor-targeting peptides with high affinity towards integrins, a large family of cell adhesion receptors abundant on tumor cells and vasculature and thus widely used in cancer targeting [40]. Since the first RGD sequence was identified in the fibrinogen protein, many analogues have been developed with enhanced affinity and stability for integrins [41]. More interesting results have been achieved using RGD-containing cyclic peptides, which after the cleavage by proteases at the tumor site produces the fragment CRGDK which contains a CendR motif (R/KXXR/K) binding to neuropilin-1 (NRP-1) receptor and inducing tissue penetration [42]. In addition, RGD-containing peptides were also effective when incorporated into different delivery systems to produce selective drug release in tumor site. In particular, the RGD motif was incorporated into PEGylated polyamidoamine (PAMAM) dendrimer Doxo (PAMAM-PEG/Doxo) conjugates [43,44] and showed similar in vitro cellular uptake and cytotoxicity to C6 glioma spheroids as those decorated with the cyclic RGD motif c(RGDyC), likely because they share the same targeting mechanism. The RGD motif was also integrated into larger peptide structures such as knottins. High proteolytical stability and selectivity for integrin receptors were granted by this highly constrained structure, and knottins were developed as imaging contrast for mouse cerebellar medulloblastoma [45]. The RGD peptide has also been conjugated to gemcitabine to obtain a strong inhibitor of brain, breast, ovarian and pancreatic cancer cell lines in vitro [46].

The NGR motif is another ligand-directed targeted delivery of different chemotherapeutics. The NGR sequence targets CD13, a tumor marker associated with the myeloid linage. The turn induced by cyclization is responsible for populating the receptor-binding conformation, enhancing affinity [47]. Recently, Dreher, et al. synthesized the cyclic peptide cKNGRE with a lactam bridge, which when conjugated to temperature-sensitive liposomes, resulted in a 3.5-fold higher affinity than the linear one [48]. 

Another important target at the tumor level is epidermal growth factor receptor (EGFR) that is overexpressed in a variety of tumors [49,50]. Furthermore, several other peptide-targeting receptors overexpressed in different tumor types can be used to decorate NVs and mediate targeted cancer therapy.

Cyclization has been widely applied to enhance activity. The cyclic heptapeptide A7R (ATWLPPR), which binds two glioma markers, VEGFR2 and NRP-1, produced valuable results [51]. Lu, et al. demonstrated that the cyclic A7R conjugating to Doxo-loaded liposomes could enhance the anti-glioblastoma effect in vivo [52]. 

Cyclic peptides are also superior BBB shuttles, enhancing selective delivery of cargoes into the brain [53]. A noticeable example is apamin (CNCKAPETALCARRCQQH), a bicyclic peptide naturally occurring in bee venom that can circumvent the BBB and deliver several cargoes [54]. Unfortunately, the bicyclic natural peptide is characterized by high neurotoxicity and immunogenicity [55].

### 2.4. CPP-Decorated Anticancer NVs

One of the most prominent issues is poor tumor penetration; in fact, nanomedicines are generally located near the tumor vessels and are unable to access the deep tumor tissue resulting in drug resistance and limited therapeutic efficacy. This serious obstacle calls for more effective drug delivery systems to exploit for the treatment of many pathologies. One opportunity to improve the control of diseases and reduce resistance issues is the use of the cell-penetrating peptides (CPPs) [56,57,58,59]. CPPs are a set of short peptides, made of up to 40 amino acids, which hold a significant capacity to cross membranes and are widely used to transfer many biologically active cargoes into the cells.

CPPs are widely employed for the delivery of a variety of drugs to treat many pathologies including cancer. In fact, the encouraging results of CPP-derived therapeutics in several tumor models demonstrated the key role played by CPPs to facilitate intracellular drug delivery to tumors whilst sparing normal healthy tissues. The power of CPPs relies in their low cytotoxicity, high efficiency and no limitations with respect to the dose or size of the cargo. Furthermore, CPPs can cross membranes at low concentrations, both in vivo and in vitro, without significant membrane damage. More than 30 CPPs are in clinical trials, but only a few have entered Phase III [60].

The prototype of cationic CPPs is TAT derived from HIV-1 proteins [61]. Cationic CPPs present excellent cytoplasmic membrane affinity due to the net positive charge, determined by the presence of arginine and lysine residues. The positive charge allows their electrostatic interaction with the cell membrane surface followed by complete internalization through a receptor-independent mechanism usually involving endocytic pathways. The nuclear localization sequences (NLS) belong to the cationic CPPs and are short peptide sequences able to enhance nucleus import through nuclear pore complex [62]. NLS are characterized by the presence of lysine, arginine and proline amino acids which are key to performing this specific function; the first identified NLS (PKKKRKV) was derived from the simian virus 40 large T antigen [63].

Amphipathic CPPs comprise both polar and non-polar regions, which may be differently distributed in the primary or secondary structures. Among amphipathic CPPs are penetratin [64] and MAP [65]. Primary amphipathic CPPs are based on the covalent binding of signal peptides or fusion peptides with a hydrophilic NLS [66]. Secondary amphipathic CPPs form α-helical or β-sheet structures, characterized by the presence of hydrophilic and hydrophobic separate faces [66,67,68]. Interestingly, some CPPs fold in a-helical or β-sheet structures when integrating into a cytoplasmic membrane; the change in the secondary structure plays an essential role in cellular uptake.

The major feature of hydrophobic/membranotropic CPPs is the presence of a large number of hydrophobic residues in the sequence, which are likely to be responsible for both a low net charge and a high cell membrane affinity [56,57,58,59,69]. Their simultaneous hydrophobic and amphipathic nature determines their high propensity for binding to lipid membranes [70]. Viral fusion peptides belong to the group of membranotropic CPPs [56]. The hydrophobic CPP family comprises K-FGF [71], C105Y [72] and gH625 [73,74,75,76,77].

It is widely accepted that CPPs can deliver a variety of cargoes into cells. Nonetheless, there is still huge disagreement about the cell uptake mechanisms. The uptake efficiency is strongly dependent on the interaction of each CPP with cellular membranes and when bound to a cargo, several other factors have to be taken into consideration such as concentration, structure, charge and length, but also the cell types (such as cell surface sugars, membrane lipid composition and peptide-to-lipid ratio) and the characteristics of the bound cargo (size, type and charge) [78]. Obviously, many other experimental factors may interfere in the mechanism of membrane translocation, such as pH and temperature. Nevertheless, it is agreed that the internalization mechanisms can be either energy independent (direct translocation) or energy dependent (endocytosis). According to many factors, several internalization routes may act simultaneously.

Cellular uptake proceeds through initial electrostatic contacts between cationic CPPs and negatively charged glycosaminoglycans and proteoglycans present on the cell surface. Subsequently, peptides enter cells by means of different transduction models which include the formation of stable pores (such as barrel-stave and toroidal pore models) or micellization in a detergent-like way (carpet model) [79]. These models also explain the translocation mechanism of some antimicrobial peptides [56,59]. Hydrophobic/membranotropic CPPs are characterized by a robust affinity for the bilayer governed by hydrophobic interactions. Compared to cationic CPPS, they can penetrate deeper into the bilayer hydrophobic core, but they are unable to span the membrane in a pore-like fashion. Likely, the ability of membranotropic CPPs to self-associate at the interface between the membrane and the aqueous compartments is fundamental for direct translocation. Likely, the direct translocation ability of membranotropic CPPs may be attributed to their capability to self-associate at the interface between the membrane bilayer and the aqueous region. The same model cannot be applied for the rationalization of the uptake mechanism of cationic CPPs which lack hydrophobic amino acids.

Furthermore, another major difficulty is represented by endosomal entrapment and the eventual escape from endosomes or lysosomes to gain access to the intracellular target sites and exert biological activity. Membranotropic CPPs uptake mechanism mainly involves direct penetration of the membranes; this model likely allows the cargo to be immediately available [56,80] and will also modify the toxicity of the internalized drug, thus favoring the overcoming of drug-resistance issues. 

For instance, Doxo has been simultaneously coupled to penetratin and Tat [81] and shown to be internalized by several drug-sensitive and drug-resistant cancer cells. Liposomes externally decorated with the CPP gH625 were also able to overcome Doxo resistance in lung adenocarcinoma cell lines [82]. 

### 2.5. Stimuli-Responsive Anticancer NVs

The strategies of stimuli-activable NVs represent a new important focus of scientific research. Several strategies have been used for delivering cargos that should reach specific sites. The development of activable NVs is a challenging approach used for target-specific delivery, particularly to cancer tissue and may exploit several features typical of cancer cells. 

Cancer cells require a high amount of energy provided by an increased level of anaerobic glycolysis and consequently a high production of lactate and protons with the result of an acidic tumor microenvironment, which contributes to tumor growth, metastasis and drug resistance. Thus, pH-responsive peptides can be bound to NVs [83]. The pH low insertion peptides (pHLIPs), [84] are able to change state according to the pH of the environment. Lower pH favors the binding of protons and enables deeper penetration into the membrane, which subsequently promotes cell internalization.

Hypoxia is characteristic of the tumor microenvironment and is widely used for targeting cancer tissues. Silica nanoparticles conjugated to the oxygen-dependent degradation domain of hypoxia-inducible factor-1a protein fused to TAT, were stable in hypoxic environments and capable of penetrating cancer cells. The cargo was released in hypoxic zone of tumors thanks to the cleavage of the hypoxia-sensitive cross-linkers [85].

Cancer environment is also characterized by an oxidative environment, which could be used for the development of redox-NVs, in fact the glutathione level inside the cancer cell is higher than in normal tissue. Bioactive compounds conjugated to NVs through disulfide bonds could be cleaved by GSH. Many H_2_O_2_-responsive NVs have been developed for tumor therapy to enhance the stability of NVs and to increase the intracellular delivery of bioactive compounds overcoming the cellular barriers. For instance, GSH induced the reduction of Fe^3+^ to Fe^2+^, increasing the delivery of siRNA and Doxo [86].

In the tumor microenvironment and cancer cells, several enzymes are upregulated and can be used to induce a modification and drug release from different nanocarriers. The extracellular membranes of cancer cells are coated with enzymes that are not expressed by healthy tissue. For instance, matrix metalloproteases are capable of degrading protein structures in the extracellular matrix and play a key role in tumor invasion and metastasis. Tsien, et al. used a cationic polyarginine CPP, shielded with a negatively charged peptide domain and connected by a peptide linker recognized by metalloproteases, to specifically deliver the drug into tumor cells [87].

Wang, et al. reported a pH and reduction-responsive polymeric lipid vesicle for the targeted delivery of Doxo [88]. Mao, et al. designed a micelle to control the release of Doxo under a high GSH tumor microenvironment [89].

Poly(N-vinylcaprolactam) (PNVCL) is a thermo-responsive polymer widely applied in drug delivery and thermo-triggered drug release systems [90]; several research groups exploited the combination of chitosan (CS) and PNVCL for cancer theranostics [91]. The phase change in the copolymer triggers drug release at the temperature of the tumor microenvironment, then acid-responsive properties of CS represent the second trigger for drug release. The group reported on the stepwise release of two co-delivered agents into tumor cells, exploiting NVs functionalized with the CPP H6R6 and loaded with Doxo and oleanolic acid [92]. The system was selectively targeting tumor cells and releasing the drugs with a significant enhancement in antitumor activity both in vitro and in vivo. Niu, et al. developed a chitosan-based cascade-responsive Doxo delivery system to overcome some hard-to-treat cancers [15] based on PNVCL-CS nanoparticles, which were further modified with a CPP and loaded with Doxo. In vivo experiments demonstrated a significant reduction in tumor volume and prolonged life span, with no obvious systemic toxicity [15].

Sun, et al. synthesized a novel pH and temperature-responsive paclitaxel-loaded drug delivery system based on chitosan and di(ethylene glycol) methyl ether methacrylate. The hyaluronic acid allowed active targeting of CD44-overexpressing human breast cancer cells [93].

Pucci, et al. developed ultrasound-responsive drug-loaded piezoelectric nanoparticles functionalized with ApoE that could be remotely activated with ultrasound-based mechanical stimulations to locally induce drug release [94].

Mesoporous organosilica nanoparticles (HMONs) of dimensions lower than 75 nm were loaded with perfluoropentane (PFP) and the photothermal agent indocyanine green (ICG) and a disulfide-containing paclitaxel (PTX) prodrug was incorporated on its surface. Both in vitro and in vivo experiments show that the nanoparticles present potent synergistic chemo-photothermal therapy [54].

Gene or oligonucleotide therapies are very promising therapeutic options to treat cancer [95,96]. siRNAs present many advantages compared to conventional chemotherapeutics such as lower toxicity, less side effects and the possibility to target any gene of interest. However, the therapeutic efficiency of RNA interference strategies is still not satisfactory because siRNAs have difficulty in crossing cellular membranes and are extremely sensitive to digestion from nucleases and to enzymatic degradation. Therefore, the encapsulation of siRNAs into smart-delivery systems represents an effective strategy to overcome their disadvantages. For example, ABCB1 siRNA was encapsulated into hyaluronic acid (HA) nanoparticles to target cancer cells overexpressing surface protein CD44 [97], while in an in vitro study, MDR1 siRNA was delivered through nanoparticles to knock down the expression of MDR1. Moreover, a promising strategy also consists of the co-delivery of siRNA and chemotherapeutic drugs to improve the efficiency of the drug [98]. Jia, et al. reported a smart carrier, liposome-coated Prussian blue @ gold nano-flower, coated with CTP for the selective delivery and controlled release of siRNA targeting the mutant gene of Kras in pancreatic tumors. The developed NV could efficiently convert NIR light into heat for gene-photothermal synergistic therapy both in vitro and in vivo [99]. Wan, et al. reported a new type of reactive oxygen species (ROS)-sensitive NV to load siRNAs, which was formed from a stimuli-responsive polymer with a poly-l-lysine-thioketal and modified cis-aconitate to facilitate endosomal escape and the CTP iRGD [100]. Jin, et al. reported a dual-loaded nano-delivery system to deliver paclitaxel (PTX) and a siRNA for breast cancer therapy in mice models [101].

Figure 3 summarizes the different strategies for obtaining stimuli-responsive anticancer NVs with peptides.

### 2.6. Theranostics as a New Approach in Cancer Treatment

The term theranostics means combining diagnosis and therapy. Image-guided therapeutics (theranostics) has stimulated the understanding of the mechanistic aspects of multiple disease-related signaling to recognize and enable easy and early diagnosis [102]. Several nanomaterials are emerging for their magnetic properties, which can be monitored by MRI (magnetic resonance imaging) and/or stimulated via external magnetic fields. In particular, superparamagnetic iron oxide nanoparticles (SPIONs) may serve as theranostic nanoplatforms to integrate diagnostical imaging and therapy (magnetic hyperthermia, photothermal therapy, controlled drug delivery and release, etc.) in vivo [103]. One of the significant advances of theranostics is also the possibility to monitor animals and to observe biological events without the need of sacrificing the animals [93]. SPIONs may serve for many other imaging modalities such as: (i) optical imaging (OI), (ii) photoacoustic imaging (PAI), (iii) computed tomography (CT) and (iv) nuclear imaging, comprising both single-photon computed tomography (SPECT) and positron emission tomography (PET). Recently, theranostic nanoplatforms were developed based on SPIONs coated with PEG. Starting from those nanoplatforms, it was possible to obtain NVs of small interfering RNA (siRNA). In order to deliver the siRNA to the advanced breast cancer, NVs were decorated with single-chain antibody fragments (scFv) of antibody specific to human epidermal growth factor receptor-2 (HER2) [104]. In contrast, to target triple-negative breast cancers, the siRNA NVs were functionalized with the CPP gH625. Their efficiency to inhibit the GFP in vitro, in MDA-MB-231 cells, was 1.7-fold higher than that of the non-targeted NVs [76].

## 3. NVs of Antimicrobial Drugs

### 3.1. NVs Applications against Infectious Diseases

Antibiotics usually exhibit limited activity to kill intracellular microbes and a high dose of the antibiotic is needed to eliminate intracellular bacteria, resulting in possible adverse effects and toxicity. Moreover, many pathogenic microorganisms con form biofilms, which allow the bacteria to adapt to environmental pressure by adjusting their metabolism and developing strong drug resistance [2]. Furthermore, due to bacterial resistance, sepsis is even more difficult to treat and how to target the infection site and effectively treat and control sepsis is still an unresolved issue. Recently, many studies involving antibacterial drug delivery NVs, based on functional materials, have been developed [30,105,106]. The NVs may act as inert drug delivery system of drugs already in clinical use, enabling loading, stability and release of the drug. The NVs may have antimicrobial activity on their own or they may act with a synergic activity both delivering other drugs and exploiting their own antimicrobial activity (Figure 4).

In NVs as drug delivery tools for antimicrobials, the drug can be adsorbed, encapsulated or bound with an on-demand release mechanism. The main aim is to improve the drug pharmacokinetics and pharmacodynamics and NVs advantages include: (i) improved antibiotic/drug solubility, (ii) controlled and sustained release of the loaded antibiotic/drug, (iii) prolonged systemic circulation and (iv) improved efficacy against intracellular infections since they can enter the host cells via endocytosis or other mechanisms as already reported for the NVs for cancer therapy.

Some NVs have also been explored to be used as antimicrobial agents on their own. Their mechanism of microbial killing is different from that of conventional antibiotics; in fact, thanks to their physicochemical properties they can enter the bacterial cell membrane bilayers and reach the cytoplasm, simultaneously disrupting the function and integrity of the membrane.

Zhang, et al. designed and developed bioresponsive NVs for drug-targeted delivery of antibiotics and anti-inflammatory drugs, to accomplish effective sepsis control and treatment [107]. To address resistance issues, Hou, et al. developed a mRNA encoding an antimicrobial peptide, an enzyme-sensitive linker peptide and a lysosomal signal protein [108] which was able to carry and release into the cytoplasm the mRNA. Upon macrophage interaction, bacteria were first wrapped in the phagosome, and then the phagosome fused with the lysosome. The antimicrobial peptide was released into the lysosome, effectively reducing the number of drug-resistant bacteria and drastically improving the host survival rate.

S-thanatin (Ts) is an AMP, which was coupled onto the surface of a liposome and exploited for bot targeting and antibacterial activity [109]; the liposomes were further loaded with the antibiotic levofloxacin. The obtained compound significantly enhanced the antibiotic internalization, resulting in a synergistic effect. In vivo, the bacteria were rapidly cleared, decreasing the lethality rate of septic shock.

Graphene oxide (GO), a single layer with a two-dimensional honeycomb lattice structure made of carbon atoms, is a promising engineered NV with strong antibacterial effects and was highly studied for its mechanisms of bacterial killing [110].

Dendrimers inhibit microbial pathogens but also play a key role as drug delivery tools [111]. Cationic polyamidoamine (PAMAM) dendrimers, before being studied as active antimicrobials, were investigated as drug carriers of conventional antibiotics [111,112]. Their antimicrobial activity is associated with multivalency, the presence on the dendrimer surface of more moieties capable of destroying the pathogen membrane. In fact, high generation of cationic dendrimers shown a high biocide activity [113]. Dendrimers functionalized with tetra or octapetides have shown antibacterial activity against Gram-positive and Gram-negative bacteria [114,115,116].

Hydrogels are cross-linked polymeric networks that exhibit antimicrobial activity, or when loaded with antimicrobial agents, can be coated on urinary catheters, contact lenses or used in wound healing [117]. Yeo, et al. have designed and developed a hydrogel made of polyethylene glycol dimethacrylate linked with a polyethylenimine star copolymer [118], that has an intrinsic antimicrobial activity able to eradicate biofilms of methicillin-resistant *Staphylococcus aureus*, carbapenem-resistant *Pseudomonas aeruginosa* and *Acinetobacter baumannii.* More interesting results are obtained when an AMP is attached to a hydrogel and delivered, because the peptide drugability, including improved permeability, low proteolysis, and long half-life, is improved. Atefyekta, et al. showed that when the AMP of sequence RRPRPRPRPWWWW was linked to Pluronic F127 hydrogel exhibited strong activity against multidrug-resistant bacteria (methicillin-resistant *S. aureus* and multidrug-resistant *Escherichia coli*) for up 24 h [119]. In addition, the serum stability of AMP increased significantly when it is attached to hydrogel, showing stability of 48 h and preserving 50% of its antimicrobial activity.

### 3.2. NV Potential against Biofilms

Many conventional antibiotics fail to treat infections due to the formation of biofilms by pathogenic microorganisms [120]. The rigid structure of biofilms formed by extracellular polymeric substances (EPS) creates a barrier against the penetration of antimicrobial agents [121]. Most antimicrobial agents are designed to treat planktonic pathogens and result ineffective at treating biofilm infections [122,123]. The development of new therapeutic strategies against biofilms, especially for drug-resistant bacteria/fungi, is critically necessary. The current treatment for biofilm infections is the use of combined antibiotics [124], with consequent toxicity and increased bacterial resistance. Great interest is thus devoted to the search of novel biofilm infection treatments. Several studies are aimed at the use of delivery systems to eradicate or treat the biofilm infections, including nanoparticles, dendrimers and nanofibers.

NVs can be effectively used for the delivery of antibiotics but also of other molecules, such as enzymes and essential oils [125,126]. In fact, most antibacterial agents have difficulty in penetrating through the EPS matrix produced by the biofilm [127]. A valuable strategy to enhance and prolong the antimicrobial efficacy of antibiotics is their entrapment in NVs. The interactions between the biofilm and NVs involve three steps: (i) NV transfer close to the biofilm, (ii) attachment to the biofilm surface and (iii) location into the biofilm [128]. The initial interactions between the biofilm and NVs are mainly determined by the electrostatic force; in fact, the negatively charged matrix easily interacts with the positively charged NV [129]. Following this initial electrostatic interaction, NVs distribute and diffuse into the biofilm through the EPS matrix and can kill pathogens via protein function inhibition, DNA damage, translation disturbance and/or transcription dysregulation [130].

Peptides containing arginines and tryptophans were bound to dendrimers and tested against planktonic persister cells, regular planktonic cells and persister cells in preformed *E. coli* biofilmst [131,132]. A cationic carbosilane dendrimer functionalized with an antibiotic and a peptide was developed to inhibit the formation of *S. aureus* biofilms, showing both antibiofilm-damaging and -inhibitory activities [133].

The presence of AMPs on the surface of NVs increases the effective local concentration of the peptide compared to soluble analogues and is the driving force for improved antibacterial activity. A novel versatile peptide biomaterial was obtained using self-assembling peptides, and was framed on its external surface with WMR, an analogue of the marine AMP myxinidin [19,134]. The NV showed increased stability and half-life and the multivalent presentation of WMR on its surface improved antibiofilm activity against the Gram-negative bacterium P. aeruginosa and the fungus Candida albicans [135,136,137] representing a sound strategy to design smart materials, which may also contain a conventional antibiotic and be stimuli responsive (pH-driven), releasing the loaded antibiotic and antimicrobial peptide following a change in pH.

### 3.3. NV Potential as Vaccines

Peptide-based vaccines represent promising candidates in the prevention of infectious diseases. It is thus necessary to preserve in vivo peptide stability and immunogenicity to achieve protective immunity. A peptide-based vaccine delivery system based on dendrimers was recently developed against *Chlamydia trachomatis* [138]. The carrier isa PAMAM dendrimer bound to an AMP (AFPQFRSATLLL) and was able to induce Chlamydia-specific serum antibodies after subcutaneous immunizations. The authors clearly demonstrated that dendrimers constitute a promising platform for the delivery of peptide vaccines.

### 3.4. NV Potential as Antiviral Compounds

Drug delivery systems are also fundamental against viral pathologies. In fact, several dendrimer-based systems possess antiviral activities being able to either prevent binding of viruses to the target cell surface or prevent replication of the viral genome. 

Efficient antiviral peptides have been developed, which were tagged with cholesterol being able to self-assemble into nanoparticles in water solution until they reach the target/viral membranes where they are integrated. An optimal balance between self-assembly and membrane integration regulates activity [139].

Dendrimers functionalized with sialic acid comparable in size to influenza virus were proved successful in blocking viral interactions [140]. A Janus-like dendrimer functionalized with peptides derived from *Herpes Simplex Virus Type 1* (HSV-1) glycoproteins showed promising inhibitory activity [74,141]. This versatile platform enhances viral inhibition by multivalent binding and interactions [13]. This study brought the development of a platform for the delivery of antiviral peptides which proved highly effective against HSV-1 and the idea of selecting peptides, that play different roles in the complex antiviral mechanism and combine them on NVs, represents a novel strategy to obtain smart antiviral drugs to fight many worldwide threatening viral infections [13].

## 4. Commercial NV Drugs

Several nanodrugs for the treatment of bacterial and viral infections are on the market. Amphotericin B has strong antibacterial activity on Candida, Aspergillus and fungi, but is associated with several drawbacks, such as nephrotoxicity, low solubility and low bioavailability (<0.9%) [142]. The use of liposomal amphotericin B (AmBisome^®^; Abelcet^®^) improves the circulation time; moreover, the nanoformulations based on amphotericin B improve the tolerated doses without decreasing the therapeutic effect of amphotericin B [143].

VivaGel^®^ is another peptide dendrimer in clinical use, which has been approved and marketed by Starpharma (NCT01577537, Table 3). It is used as a mucoadhesive gel for the treatment and prevention of BV (bacterial vaginosis) and as a gel for the prevention of sexually transmitted infections (NCT00331032, IST), including HIV, HPV and HSV-2. It is also commercially available under the LifeStyles^®^DualProtect ™ brand name as a condom lubricant in Japan, Australia and Canada. It has been approved in Australia and Europe for BV and its approval in US 587 and other countries is pending [144].

Regarding the drug delivery in cancer cells, numerous nanovectors were approved by FDA, while most of them are currently in clinical trials [145]. In 2017, FDA-approved liposomes VYXEOS for the treatment of acute myeloid leukemia (NCT01696084), in which cytarabine and daunorubicin are encapsulated to have a synergistic effect. Since 2016, numerous clinical trials of VYXEOS have been carried out and the most recent trial has regarded a wide patient population including children. Additionally, different kinds of NVs have been approved by the FDA. In particular, the NBTXR3/Hensify hafnium oxide nanoparticle displays increasing external radiotherapy thanks to the presence of hafnium. This delivery system has been approved for the local treatment of tissue sarcoma and has also been expanded for the treatment of prostate cancer (NCT02805894) and lung cancer (NCT03589339).

## 5. Conclusions

The challenge of the future for most pathologies is the combination therapy. This strategy may include the combination of conventional drugs or novel drugs with NVs able to simultaneously co-deliver compounds with hugely distinct characteristics. Despite the recent significant progress in the development of NVs and recognition of their potential, scientific and technological gaps continue to impede their translation from bench to bedside, and just a few nanosystems have undergone clinical trials.

The search for personalized medicine has enhanced interest in peptide-based approaches. As a matter of fact, peptides, being at the interface between naturally occurring bioactive molecules and chemically synthesized ingredients, may play different roles and are promising molecules for the treatment and control of a range of disease states including cancer and infectious diseases. Compared to traditional drugs, peptides present several advantages: (i) high biocompatibility, (ii) high selectivity for targets, (iii) low toxicity, (iv) easy elimination from the body, (v) increased therapeutic efficacy and (vi) lower capacity for side effects. Regardless of this, their widespread application has been limited by (i) their low half-life, (ii) their low oral bioavailability and (iii) their limited intestinal permeability linked to polarity and molecular weight. Furthermore, the discrepancy between in vitro and in vivo efficacy prevents clinical translation. Most peptides approved for clinical utilization are designed for topical use, which avoids unpredicted systemic toxicity, increases local effective concentration and decreases the chances of degradation. Despite these challenges, researchers and pharmaceutical industries continue to collaborate to clinically translate peptides of therapeutic value.

A challenging strategy to overcome peptide limits is represented by the application of nanotechnology and the design of peptide NVs to facilitate the absorption and distribution of the drug, which can circumvent the enzymatic activity, allowing the exploitation of their enormous therapeutic potential. Indeed, the incorporation of peptides into artificial materials has become an effective approach to improve the surface properties of materials for many applications and overcome the limits of peptide drugs.

NVs offer the possibility to expose a highly multivalent surface to warrant maximum interaction with surroundings. Additionally, NVs may carry different active molecules to improve the activity and potency and to endow the NV with additional activities providing the opportunity to design smart materials.

For instance, the incorporation of AMPs into NVs has become an effective strategy to improve the surface properties of materials for many applications while reducing drawbacks such as hemolysis, allergic responses and development of resistance. Particularly attractive is the possibility of establishing an innovative design principle through the combination of AMPs and conventional antibiotics on the same NV, which may open new avenues in reducing both the administration dose of the antibiotic and the development of resistance and in repurposing already approved drugs. For instance, the food industry is greatly fascinated by the development of antimicrobial nanomaterials that avoid the development of resistance without acting on specific targets. Useless to say that the development of peptide NVs for the delivery across the BBB of AMPs and antibiotics is also a key approach to treat brain infections. Due to the growing problem of drug resistance, the design of antiviral drugs represents a turning point in the global fight against viruses. By modulating the NV-coupled peptides, one could move from one infection to another infection quickly and this is a great advantage when unexpected infections occur.

Certainly, peptides in cancer are not a new idea and new CTP and natural/synthetic cytotoxic peptides are continuously discovered. Combining several therapy cocktails will likely turn out to be effective against cancer and represents the most reasonable strategy to produce the magic bullet that specifically targets and eradicates tumor cells. Furthermore, the combination of AMPs and chemotherapeutic drugs on NVs may represent an appealing strategy for cancer.

Although peptide-based NVs represent a valuable strategy to achieve personalized therapies, several limitations are still hindering their clinical translation. The main problems include the complexity of the nanosystems which makes it difficult to predict their complex effects, their production, standardization and upscaling. Nonetheless, the application of peptide NVs will help to further re-engineer thousands of natural and synthetic drugs, boosting their therapeutic potential. Future research needs to take measures for the effective clinical translation of peptide NVs and future medicine will most likely be a cocktail of NVs functionalized with different peptides and drugs which may be combined according to the pathology and the patient to be treated.

## Figures and Tables

**Figure 1 pharmaceutics-14-01235-f001:**
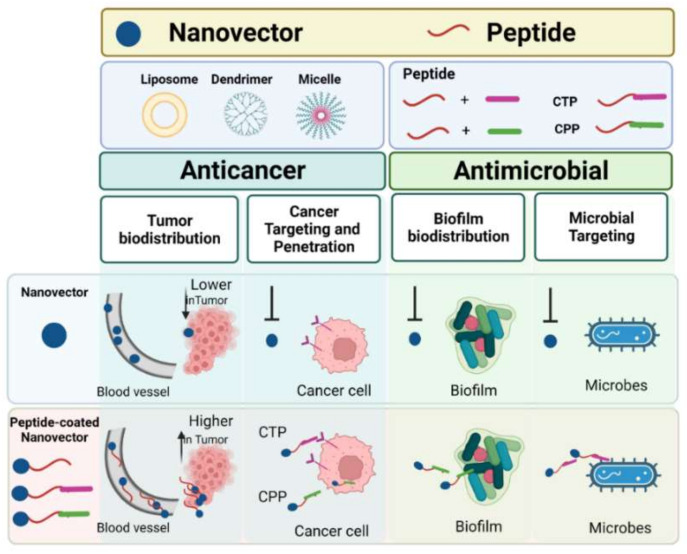
The main advantages in drug delivery by peptide-coated NVs with cell-targeting peptides (CTP) or cell-penetrating peptides (CPP) in comparison to NVs alone are reported with particular attention to their anticancer (on the **left**) and antimicrobial (on the **right**) applications. This figure was created with BioRender.com (accessed on 2 February 2022).

**Figure 2 pharmaceutics-14-01235-f002:**
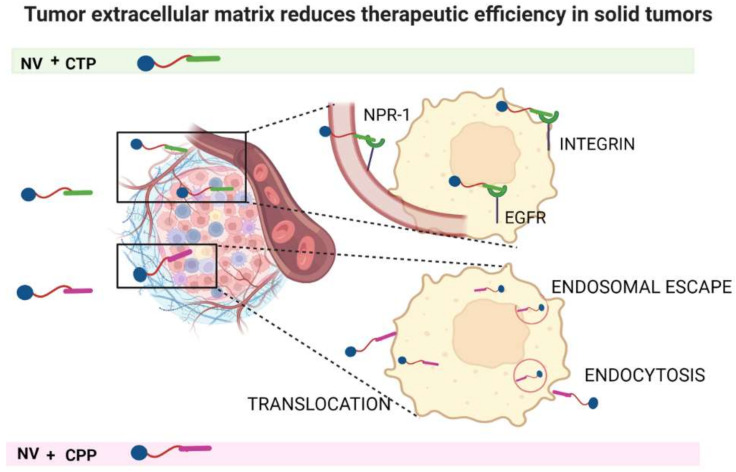
The representation of proposed mechanisms of action of CTP- and CPP-decorated NVs involved in selective drug delivery in cancer cells. CTP exploit overexpressed receptors to selectively target cancer cells; CPP enhance internalization through different mechanisms such as endocytosis, translocation or endocytosis followed by endosomal escape. This figure was created with BioRender.com (accessed on 2 February 2022).

**Figure 3 pharmaceutics-14-01235-f003:**
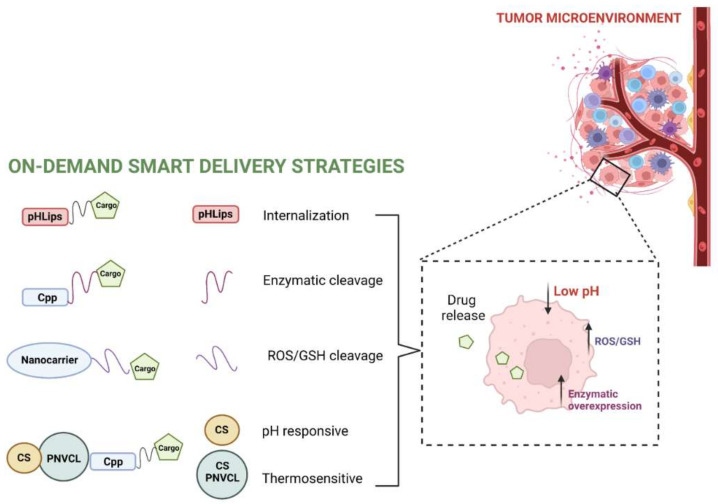
Illustration of different on-demand delivery strategies exploited for achieving a stimuli-responsive drug release from peptide-based NVs in the tumor environment. This figure was created with BioRender.com (accessed on 2 February 2022).

**Figure 4 pharmaceutics-14-01235-f004:**
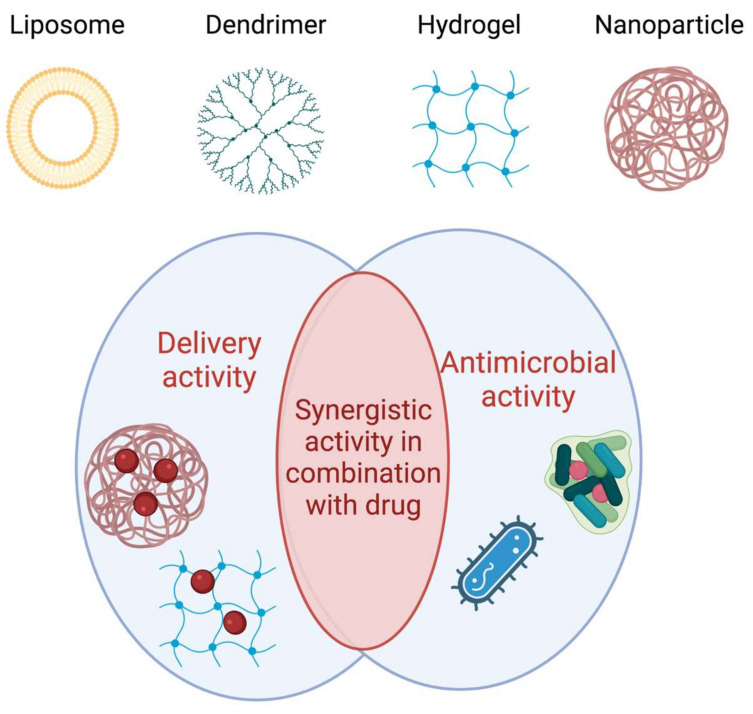
Illustration of main functions of some representative NVs in antimicrobial applications. NVs have both antimicrobial and delivery activities and may act with a synergistic activity in combination with other drugs. This figure was created with BioRender.com (accessed on 2 February 2022).

**Table 1 pharmaceutics-14-01235-t001:** Classification of the most common nanovectors, their structures and properties.

Nanovectors	Nature	Size (Diameter)	Technique Used	Application	Advantage	Ref.
Liposomes	Lipidic	50–200 nm	DLS	Anticancer Antimicrobial	Biocompatibility Biodegradability Functionalization Solubility	[7,8]
Micelles	Lipidic	10–100 nm	DLS	Anticancer Antimicrobial	Penetration Cellular uptake Permeability	[9,10]
Mesoporous Silica Nanoparticles	Synthetic	2–50 nm	SEM-TEM	Anticancer Antimicrobial	Functionalization Biocompatibility	[11,12]
Dendrimers	Synthetic	1–15 nm	DLS	Anticancer Antimicrobial	Biocompatibility Solubility Polyvalency Functionalization	[13,14]
Chitosan Nanoparticles	Synthetic	10–200 nm	DLS	Anticancer Antimicrobial	Amphipathicity Biocompatibility Biodegradability	[15,16]
PLGA Nanoparticles	Synthetic	20–200 nm	TEM	Anticancer	Biocompatibility	[17,18]
Peptide-based Nanocarriers	Synthetic	50–200 nm	TEM	Anticancer Antimicrobial	Targeting capability Biocompatibility	[19]

**Table 2 pharmaceutics-14-01235-t002:** The most popular methodologies used for covalent coupling.

Peptide Functional Groups	NV Functional Group	Advantages	Disadvantages
NCS	NH_2_ or SH	Stable linkages, selective reaction, good yield	
NHS-esters			
Maleimide			
SH	SH, maleimide, bromoacetylation, vinylsulfone		
NH_2_	COOH	Stable linkages, good yield	The presence of more reactive groups in peptide sequence
N_3_	Alkyne	Stable linkages, selective reaction, good yield	Requires catalyst

**Table 3 pharmaceutics-14-01235-t003:** Examples of commercial NVs evaluated in clinical trials.

Collaborator	Commercial Name	Nanovector	Medical Application	Clinical Trials Identifier (NCT#)	Period of Clinical Trial
M.D. Anderson Cancer Center	AmBisome®	Liposome	Fungal infections	NCT00418951	01/2007–08/2012
Starpharma Pty Ltd.	VivaGel®	Dendrimer	Bacterial vaginosis	NCT01577537	04/2012–07/2019
National Institute of Allergy and Infectious Diseases	VivaGel®	Dendrimer	Sexually transmitted infections	NCT00331032	05/2006–08/2013
The Leukemia and Lymphoma Society	VYXEOS	Liposome	Acute myeloid leukemia	NCT01696084	09/2012–09/2020
Nanobiotix	NBTXR3/Hensify	Nanoparticle	Prostate cancer	NCT02805894	06/2016–05/2021
Nanobiotix	NBTXR3/Hensify	Nanoparticle	Lung cancer	NCT03589339	08/2018–active

## Data Availability

Not applicable.

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
