# Peer review of "Peptides to Overcome the Limitations of Current Anticancer and Antimicrobial Nanotherapies"

_pharmaceutics, 2022, doi:10.3390/pharmaceutics14061235_

Round 1
Reviewer 1 Report
This review is well written and organised, however needs some minor changes to be done before its consideration to publication.
Comments
- The abstract needs revision as due to lengthy sentence it is not easy to understand.
- In table 1. minor corrections have to be made for instance, make a separate column for reference and techniques used for determination of size.
- Recent references appeared in 2022 should be cited in this area.
- The figure legends should be explained to make them more understandable and informative.
- Referencing format should be in same style.
Author Response
This review is well written and organised, however needs some minor changes to be done before its consideration to publication.
- The abstract needs revision as due to lengthy sentence it is not easy to understand.
We thank the reviewer for his/her comment and we thoroughly revised the abstract
- In table 1. minor corrections have to be made for instance, make a separate column for reference and techniques used for determination of size.
We modified accordingly
- Recent references appeared in 2022 should be cited in this area.
We now revised the reference section and there are 27 references for 2021-2022.
- The figure legends should be explained to make them more understandable and informative.
We thank the reviewer for this comment and we revised all legends.
- Referencing format should be in same style.
We revised the reference section
Reviewer 2 Report
This is a well-organized and well-illustrated paper, has an important clinical message, and should be of great interest to the readers. The review focused on the recent developments of peptide based nanovectors for anticancer and antimicrobial applications. Paragraphing is concise and good, and the article consists of major recent advancements in the field of nanobased peptide drug delivery systems for antimicrobial and anticancer use and deserve publication after some revisions listed below.
- I sugges the authors to include a small section describing the methodologies used to immobilize peptide moieties on nanoparticles, limitations of each stratergy and advantages . If possible please tabulate these data so that it would be easy for the readers to go through the recent advancements in this field.
- What factors will influence the stability or retention of peptide activity after binding to the nano vector and what measures must one take to prevent the loss of activity. Please elaborate this in the manuscript.
-
There is always a dilemma on how to conclude a review article. Since the authors have deliberately summarized huge amounts of published results, it will go a long way. It would be helpful if they can provide their own thoughts that would in turn help in finding the areas that need to be addressed. For example, what are the factors that one needs to consider while choosing a nano carrier for peptide delivery, w and what are the steps required for the fast transition of these peptide based nanomaterials for industrial scale up. In general, what measures need to be taken for the effective clinical translation of peptide nano formulations? why there are only a handful of FDA approved formulations? What limitations are hindering their clinical translation and in what direction does the future research need to be, to make the clinical translation possible? These points need to be discussed in the conclusion section.
Author Response
Comments and Suggestions for Authors
This is a well-organized and well-illustrated paper, has an important clinical message, and should be of great interest to the readers. The review focused on the recent developments of peptide based nanovectors for anticancer and antimicrobial applications. Paragraphing is concise and good, and the article consists of major recent advancements in the field of nanobased peptide drug delivery systems for antimicrobial and anticancer use and deserve publication after some revisions listed below.
- I suggest the authors to include a small section describing the methodologies used to immobilize peptide moieties on nanoparticles, limitations of each strategy and advantages. If possible please tabulate these data so that it would be easy for the readers to go through the recent advancements in this field.
We now modified the text and add a paragraph in section 2 and also a table (Table 2)
- What factors will influence the stability or retention of peptide activity after binding to the nano vector and what measures must one take to prevent the loss of activity. Please elaborate this in the manuscript.
We now added Paragraph 2.2, including this concept
- There is always a dilemma on how to conclude a review article. Since the authors have deliberately summarized huge amounts of published results, it will go a long way. It would be helpful if they can provide their own thoughts that would in turn help in finding the areas that need to be addressed. For example, what are the factors that one needs to consider while choosing a nano carrier for peptide delivery, w and what are the steps required for the fast transition of these peptide based nanomaterials for industrial scale up. In general, what measures need to be taken for the effective clinical translation of peptide nano formulations? why there are only a handful of FDA approved formulations? What limitations are hindering their clinical translation and in what direction does the future research need to be, to make the clinical translation possible? These points need to be discussed in the conclusion section.
We thank the reviewer for his/her suggestion and indeed we added a new conclusion section
Reviewer 3 Report
The authors of the article titled “Peptide-based nanovectors to overcome limitations of anti-cancer and antimicrobial therapies” present the necessity of peptide based nanovectors over traditional liposomal or polymeric vehicles for drug delivery/ antimicrobial applications. The compilation is presented in a lucid manner and is of interest to a broad audience. However, the lack of clarity and information at certain areas preclude the publication of the manuscript in the current form. The major points have been discussed below:
1. Throughout the manuscript, there is an emphasis on the applications of peptide nanovectors. There should be a discussion on the shortcomings of conventional therapeutics at a greater length. In Line 42-45, there is only a brief description of the side-effects of chemotherapy. There is no mention of which therapy has been alluded.
2. In Table 1, the shape column should be eliminated. All the different nanoparticles can be arranged in different shapes. Also, in the last row, the mention of peptide based nanocarriers as “on-demand nanostructures” is vague and should be re-worded.
3. In Line 46-62, polymeric and anisotropic nanoparticles have been used interchangeably. This should be discussed with more clarity.
4. In Line 75-96, there is a mention of role of PEGylation in overcoming challenges of clearance. This should be discussed in the context of peptide based nanocarriers for the purpose of this review.
5. Section 2.2 and Section 2.3 should be divided into subsections for better clarity.
6. In Lines 282, penetratin has been introduced for the first time in the article. This should be elaborated before mentioning drug delivery capacity. The words “Doxo” and “Dox” has been used as the acronym for Doxorubicin. This should be used consistently through the article, preferably Dox.
7. There should be a mention of inorganic nanoparticles coated with peptides used for drug delivery
8. There should be a detailed description of stimuli responsive gene delivery with peptide nanovectors.
9. There should a commentary on repurposing of CPPs and Antimicrobial peptides.
10. Many references are incomplete where there’s no mention of the journal name or other details. This should be checked thoroughly and revised.
Author Response
Comments and Suggestions for Authors
The authors of the article titled “Peptide-based nanovectors to overcome limitations of anti-cancer and antimicrobial therapies” present the necessity of peptide based nanovectors over traditional liposomal or polymeric vehicles for drug delivery/ antimicrobial applications. The compilation is presented in a lucid manner and is of interest to a broad audience. However, the lack of clarity and information at certain areas preclude the publication of the manuscript in the current form. The major points have been discussed below:
- Throughout the manuscript, there is an emphasis on the applications of peptide nanovectors. There should be a discussion on the shortcomings of conventional therapeutics at a greater length. In Line 42-45, there is only a brief description of the side-effects of chemotherapy. There is no mention of which therapy has been alluded.
We thank the referee for his/her comment and we now modified the text accordingly.
- In Table 1, the shape column should be eliminated. All the different nanoparticles can be arranged in different shapes. Also, in the last row, the mention of peptide based nanocarriers as “on-demand nanostructures” is vague and should be re-worded.
We modified the Table 1 as suggested
- In Line 46-62, polymeric and anisotropic nanoparticles have been used interchangeably. This should be discussed with more clarity.
We modified as suggested
- In Line 75-96, there is a mention of role of PEGylation in overcoming challenges of clearance. This should be discussed in the context of peptide based nanocarriers for the purpose of this review.
We mention Pegylation in the context of peptide NVs but we also kept this paragraph in the introduction as it refers to both cancer and antimicrobials.
- Section 2.2 and Section 2.3 should be divided into subsections for better clarity.
We now revised Section 2 and thus it should be more clear.
- In Lines 282, penetratin has been introduced for the first time in the article. This should be elaborated before mentioning drug delivery capacity.
We thank the referee and we added the penetratin in the group of amphiphatic CPP with a reference.
- The words “Doxo” and “Dox” has been used as the acronym for Doxorubicin. This should be used consistently through the article, preferably Dox.
We modified accordingly
- There should be a mention of inorganic nanoparticles coated with peptides used for drug delivery
We thank the reviewer for the comment, we summarized organic, inorganic and hybrid nanoparticles coated with peptides (For example see table 1, line 454 and so on)
- There should be a detailed description of stimuli responsive gene delivery with peptide nanovectors.
We added this section in the paragraph dealing with smart nanosystems (paragraph 2.4)
- There should a commentary on repurposing of CPPs and Antimicrobial peptides.
We now modified the Conclusion section adding also this concept.
- Many references are incomplete where there’s no mention of the journal name or other details. This should be checked thoroughly and revised.
We checked the reference section carefully.
Reviewer 4 Report
1. The content of the text is not described by a single topic.
In this manuscript, nanovectors, peptides, anticancer and antimicrobial therapies are described. Many research and review papers on the above-mentioned fields have been published in various journals. Therefore, the author tried to explain their merits by integrating these fields in order to secure the novelty of the manuscript. However, in the manuscript, the above fields are not integrated and explained, but each characteristic of the field is explained. Therefore, it is better to select only one of the peptide anticancer agents or the peptide antibacterial agent and describe it in more detail.
2. It is recommended to add the sequence or type of peptides used in nanovector to “table 1”.
3. To help understanding, it is necessary to add the scheme used in other reference papers.
Author Response
Comments and Suggestions for Authors
- The content of the text is not described by a single topic. In this manuscript, nanovectors, peptides, anticancer and antimicrobial therapies are described. Many research and review papers on the above-mentioned fields have been published in various journals. Therefore, the author tried to explain their merits by integrating these fields in order to secure the novelty of the manuscript. However, in the manuscript, the above fields are not integrated and explained, but each characteristic of the field is explained. Therefore, it is better to select only one of the peptide anticancer agents or the peptide antibacterial agent and describe it in more detail.
We thank the reviewer for his/her comment and indeed it was our intention to make a comprehensive review on anticancer and antimicrobial delivery tools. We understand and agree that we needed to better integrate the two fields and provide in mode details our thoughts in the conclusion. Thus, we revised the manuscript accordingly.
2. It is recommended to add the sequence or type of peptides used in nanovector to “table 1”.
We thank the reviewer for his comment. Indeed, in Table 1 we are describing different types of NVs and not peptides bound to the NV.
3. To help understanding, it is necessary to add the scheme used in other reference papers.
We modified and added several Tables in the paper to make the concepts more clear
Reviewer 5 Report
Please see the attached reviewers comment file and incorpoarte the required changes accordingly.
Thanks,

Author Response
Authors have written review on a relevant topic of nanocarriers based anticancer and antimicrobial therapies. However, the title needs to be amended, as it is confusing in its present form. It appears like peptide-based carriers (such as one dedicated to peptide as vectors such as BSA etc), whereas author have focused on peptides mostly used as targeting agent so please amend the title for more clarity. Also, there is a need for more robust data by adding some relevant research before considering it for publication. There’s need to be major revision before considering the manuscript for the publication. There is some typo- and grammatically mistakes in the text which should be corrected.
- The word Nano vectors can be replaced with non-viral nanocarriers.
We thank the reviewer for his suggestion and indeed we modified “nanovector” with “non-viral nanovector” the first time we refer to this concept in the abstract and in the text
- Surely need to amend the title for better clarity, presently it looks confusing more like protein based nanovectors then protein as targeting agent, so please double check and correct according to the review content. It should be better off as peptide as targeting agent for nanovectors rather.
We modified the Title as suggested
- In the line “In particular, the most recent results applicable to treat 18 two types of pathologies, cancer and microbial infections, are discussed, aiming at providing guidance in the overall design of precise nanomedicine and at highlighting the key role played by peptides in this field” please double check if possible replace the word precise nanomedicine.
We modified accordingly
- In the line “Additionally, future challenges and potential perspectives are illustrated, in the hope of accelerating the translational advance of nanomedicine” it should be advances.
We modified accordingly
- In figure 3 it is not obvious if it has been taken form some journal or its authors original schematic based on the reviewing. If its not your own work please cite the relevant reference in your figure legend according to that journal policy. Also, please check if you have authorization to put in your review article?
We thank the reviewer for his comment. Indeed the figures were all made by us using the software Biorender. We now modified the figure legends and added for all figures: This Figure was created with BioRender.com.
- In Figure 2. Supposed mechanisms of action of CTP- and CPP- decorated NVs to enhance intracellular action in specific cells. CTP exploit overexpressed receptors on target cells; CPP enhance internalization through different mechanisms: endocytosis, translocation or endocytosis followed by endosomal escape. Replace supposed word with Proposed or postulated mechanisms.
We modified accordingly
- In Figure 4 Representative nanovectors and their antimicrobial applications, please check the spelling of synergic, synergistic would be more appropriate as it is more commonly used.
We modified the figure
- Although it is mentioned in the abstract about future challenges and potential perspectives considering translational advance of nanomedicine, I don’t see a sub section on that. Please add a clear elaborated section.
We thank the referee for his/her comment and we rewrote the conclusion section elaborating this concept.
- In its present form conclusion is very vague, please specify considering the focus on peptide targeted nanovectors relevance and challenges, also what is the take away and directions.
We thank the referee and we modified the Conclusions accordingly
- Also, please add a clear table with few columns clearly demonstrating the translation status of the nanovectors in the anticancer and antimicrobial therapy. For instance, in the review https://www.mdpi.com/2072-6694/14/3/514 they have shown a table clearly mentioning the status of nanoparticle in the clinical trial. A similar table for the nanovectors will surely provide great insight.
We thank the reviewer for the suggestion and we added in the paragraph concerning clinical applications a Table.
Reviewer 6 Report
pharmaceutics-1724272 Peptide-based nanovectors to overcome limitations of anticancer and antimicrobial therapies.
The manuscript describes the applications of NVs with particular reference to those made of peptides, in two main types of pathologies, cancer and microbial (bacterial/viral) infections. Authors conclude that the challenge of the future for most pathologies is the combination therapy. The different sections that the review presents are adequate. Perhaps the chapter concerning CPPs decorated anti-cancer NVs could be further developed given its importance. The number of figures for this review is also adequate, as well as their quality and explanatory nature, although there are typographical errors in some legends (see line 394 in Figure 3). What is missing is more constructive criticism and personal opinion of the authors in some sections as well as in relation to current developments. On the other hand, there are references that clearly need to be expanded or replaced by more current ones, see for example references 6, 26, 29, 71, 77, 94 and 109.
Author Response
The manuscript describes the applications of NVs with particular reference to those made of peptides, in two main types of pathologies, cancer and microbial (bacterial/viral) infections. Authors conclude that the challenge of the future for most pathologies is the combination therapy. The different sections that the review presents are adequate. Perhaps the chapter concerning CPPs decorated anti-cancer NVs could be further developed given its importance. The number of figures for this review is also adequate, as well as their quality and explanatory nature, although there are typographical errors in some legends (see line 394 in Figure 3).
We thank the reviewer and modified accordingly.
What is missing is more constructive criticism and personal opinion of the authors in some sections as well as in relation to current developments.
We thank the reviewer for his/her comment and modified the manuscript accordingly.
On the other hand, there are references that clearly need to be expanded or replaced by more current ones, see for example references 6, 26, 29, 71, 77, 94 and 109.
We modified as suggested
Round 2
Reviewer 3 Report
The queries have been sufficiently addressed.
Reviewer 4 Report
i believe this manuscript is sufficient for publication.
Reviewer 5 Report
Thanks for addressing all the comments.